# Comparative Analysis of Microbial–Short-Chain Fatty Acids–Epithelial Transport Axis in the Rumen Ecosystem Between Tarim Wapiti (*Cervus elaphus yarkandensis*) and Karakul Sheep (*Ovis aries*)

**DOI:** 10.3390/microorganisms13051111

**Published:** 2025-05-12

**Authors:** Jianzhi Huang, Yueyun Sheng, Xiaowei Jia, Wenxi Qian, Zhipeng Li

**Affiliations:** 1College of Animal Science and Technology, Tarim University, Alar 843300, China; huangjianzhi2021@126.com (J.H.); 15999220945@163.com (Y.S.); huozhe211@163.com (X.J.); 2Key Laboratory of Tarim Animal Husbandry Science and Technology, Xinjiang Production and Construction Group, Alar 843300, China; 3College of Animal Science and Technology, Jilin Agricultural University, Changchun 130118, China; 4Key Laboratory of Animal Production, Product Quality and Security, Ministry of Education, Jilin Agricultural University, Changchun 130118, China; 5Jilin Provincial Engineering Research Center for Efficient Breeding and Product Development of Sika Deer, Changchun 130118, China

**Keywords:** microbe–host interactions, species differences

## Abstract

Under long-term ecological stress, the Tarim wapiti (*Cervus elaphus yarkandensis*) has evolved unique adaptations in digestive physiology and energy metabolism. A multi-omics comparison of three Tarim wapiti and five Karakul sheep was used to examine the synergistic mechanism between rumen bacteria, short-chain fatty acids, and host epithelial regulation in order to clarify the mechanism of high roughage digestion efficiency in Tarim wapiti. Metagenomic sequencing (Illumina NovaSeq 6000) and gas chromatography revealed that Tarim wapiti exhibited significantly higher acetate and total VFA (TVFA) concentrations compared to Karakul sheep (*p* < 0.01), accompanied by lower ruminal pH and propionate levels. Core microbiota analysis identified *Bacteroidetes* (relative abundance: 52.3% vs. 48.1%), *Prevotellaceae* (22.7% vs. 19.4%), and *Prevotella* (18.9% vs. 15.6%) as dominant taxa in both species, with significant enrichment of *Bacteroidetes* in wapiti (*p* < 0.01). Functional annotation (PICRUSt2) demonstrated enhanced glycan biosynthesis (KEGG ko00511), DNA replication/repair (ko03430), and glycoside hydrolases (*GH20*, *GH33*, *GH92*, *GH97*) in wapiti (FDR < 0.05). Transcriptomic profiling (RNA-Seq) of rumen epithelium showed upregulated expression of SCFA transporters (*PAT1*: 2.1-fold, *DRA*: 1.8-fold, *AE2*: 2.3-fold; *p* < 0.01) and pH regulators (*Na^+^/K^+^ ATPase*: 1.7-fold; *p* < 0.05) in wapiti. Integrated analysis revealed coordinated microbial–host interactions through three key modules: (1) *Bacteroidetes*-driven polysaccharide degradation, (2) *GHs*-mediated fiber fermentation, and (3) epithelial transporters facilitating short-chain fatty acids absorption. These evolutionary adaptations, particularly the Bacteroidetes–short-chain fatty acids–transporter axis, likely underpin the wapiti’s superior roughage utilization efficiency, providing molecular insights for improving ruminant feeding strategies in an arid environment.

## 1. Introduction

The Tarim wapiti (*Cervus elaphus yarkandensis*), a subspecies of red deer endemic to China, inhabits the arid Tarim Basin of Xinjiang Uygur Autonomous Region, where it thrives along the Tarim River and its tributaries—a rare adaptation to desertified environments [1]. Previously, compared to other ruminants (cattle and sheep) in the same geographic area, Tarim wapiti had significant advantages in terms of digestibility of dietary crude protein (CP), neutral detergent fiber (NDF), acid detergent fiber (ADF), and organic matter (OM) [2], as well as microbial protein (MCP) and acetate content in the rumen [2,3], and Tarim wapiti had unique advantages in terms of bacteria [2,3]. It might be that the Tarim wapiti’s adaptation to its distinct natural ecological habitat resulted in the development of unique digestive physiological features. Surprisingly, the digestibility of dry matter (DM), OM, CP, and crude fiber (CF) in deer was much lower than that in cattle or sheep in the same geographic region [4,5,6]. This emphasizes the roughage endurance of Tarim wapiti in desertified habitats. Karakul sheep are also an exceptional native breed in the same region, capable of adapting to desert and semi-desert environments and possessing great adaptation and resilience to hard feeding.

Previously, during the exploration of roughage tolerance characteristics of Tarim wapiti, the gastrointestinal microorganisms [7] and the overall structural composition of the rumen [8] showed unique characteristics, and compared with cattle and sheep in the same geographic area, the Tarim wapiti had an advantage in feeding and ruminating pattern [2], rate of coelomic efflux [2], and rumen bacteria [2,3]. Furthermore, rumen bacteria have a direct impact on host digestive efficiency, with their microbial community playing an important role in breaking down difficult-to-degrade fibers into absorbable tiny molecules [9]. The composition of these bacteria is influenced by a number of variables, including host genetic traits [10].

Moderately inherited rumen microbes have been found to be associated with host feed efficiency and rumen short-chain fatty acids content [11], whereas CH_4_ emissions may not be regulated through genetic effects of the rumen microbiome [12], and there are differences in the relative abundance of bacterial populations between species [3,13,14], as well as differences in protozoan abundance [4,15], while microbial differences result in inter-species significant differences between the abundance of CAZymes and KEGG direct homologs [14], which in turn lead to differences in nutrient digestion and metabolism in the organism. Furthermore, the rumen epithelium is a unique site of interaction between rumen microorganisms and their hosts, and it plays an important role in the body’s nutrient and energy cycling, as well as the organism’s health [16]. Specifically, the rumen epithelium’s uptake of short-chain fatty acids keeps the rumen pH in a suitable range, preventing rumen flora disorders caused by both excessively low and high levels [17]. In contrast to prior pure culture and single microbial species determinations, macrogenomic methods can directly assess the community function and genetic composition of the whole range of microorganisms found in environmental samples.

This study employed multi-omics analysis of three Tarim wapiti and five Karakul sheep to elucidate the synergistic mechanism among rumen microbiota, short-chain fatty acids, and host epithelial regulation underlying their high roughage digestion efficiency. Furthermore, our findings establish a theoretical foundation for optimizing roughage utilization and dietary formulation in Tarim wapiti.

## 2. Materials and Methods

### 2.1. Animal and Feeding Management

For this study, three healthy adult male Tarim wapiti (200 ± 5.0 kg, 5 years old) were selected from the 31st Regiment Field of Xinjiang Production and Construction Corps, China. The experiment was conducted on 20 December 2022 in a study area (86°45′–87°00′ E, 40°49′–40°59′ N; elevation: 820–1100 m) characterized by a continental arid climate. Karakul sheep are also an exceptional native breed in the same region, capable of adapting to desert and semi-desert environments and possessing great adaptation and resilience to hard feeding. Five Karakul sheep (18.0 ± 0.3 kg, 3 months old) were kept in a single enclosure and given the same feed formulation until they reached adulthood (50 kg). The same ration formula was used and the rations were formulated according to *Deer Production Science* [18] and *Animal Nutritional Parameters and Feeding Standards* [19]. The ratio of concentrate to roughage in the baseline diet was 3:7, and Table 1 shows the diet composition and nutritional levels. Feed twice a day, at 9 a.m. and 7 p.m., to ensure that food and water are readily available. Maintain sufficient ventilation, cleanliness, and sterilization. 

Feed concentrations of crude protein (CP), calcium (Ca), and phosphorus (P) were quantified using AOAC standard methods [20]. CP content was measured via Kjeldahl nitrogen analysis, Ca through potassium permanganate titration, and P by absorbance photometry.

### 2.2. Collection of Samples

At the end of the formal trials, carotid artery bloodletting was performed 2 h after morning eating. After opening the abdomen cavity and using the procedure described in *Anatomy of Livestock and Poultry* [21] to separate the rumen, the rumen was laid flat on a clean tray and sterile gloves were used to collect rumen fluid from the rumen ventral sac, filtered through four layers of sterile gauze, and pH was determined immediately, i.e., it was kept in a freezing tube in liquid nitrogen for fast freezing, and then stored in a −80 °C refrigerator. The rumen was dissected, and the abdominal sac tissue block was collected, washed three times in iced phosphate-buffered salt solution, the rumen epithelium was bluntly separated from the rumen musculature, sheared, and stored in a 2 mL freezing tube in a liquid nitrogen tank for subsequent qRT-PCR. Beijing Novogene Bioinformatics Technology Co., Ltd. (Beijing, China) performed macrogenome sequencing and qRT-PCR analysis.

### 2.3. Analysis of Rumen Fermentation Parameters

Rumen fluid pH was measured with a portable pH meter (FE28, Mettler Toledo, China). Short-chain fatty acids (SCFAs; acetate, propionate, butyrate, and valerate) were quantified by gas chromatography (SP7800, Beijing Jingke Ruida, China) under the following conditions: column temperature 120 °C, injector 230 °C, detector 250 °C, with 2 μL injection volume.

### 2.4. Metagenomic Sequencing

#### 2.4.1. DNA Extraction for Metagenomic Sequencing

The Magnetic Bead Method Soil and Fecal Genomic DNA Extraction Kit (catalog no.: DP712, TianGen Biotech Co., Ltd., Beijing, China) was used to extract macrogenomic DNA, 0.6% agarose gel electrophoresis was used to evaluate DNA purity and integrity, and DNA was quantified using the Qubit^®^ dsDNAHS Analysis Kit (catalog no.: Q32854, Thermo Scientific, Waltham, MA, USA).

Fecal DNA has been extracted with the MagaZorb DNA Mini-Prep Kit (catalog no.: MB2004, Promega, Madison, WI, USA). The extracted DNA’s quality was tested using agarose gel electrophoresis on a 0.6% (*w*/*v*) gel and a spectrophotometer (optical density at a ratio of 260 nm/280 nm), and the quantity of DNA was quantified using a Qubit 2.0 fluorometer.

#### 2.4.2. Macrogenomic Library Preparation and Sequencing

The Illumina libraries were generated using Illumina’s NEBnext Ultra II DNA Library Prep Kit (New England BioLabs, Ipswich, MA, USA), which included end repair, A-tail addition, sequencing junction addition, purification, and PCR amplification. After the library passed the test, Illumina PE150 sequencing was performed.

The SMRTbell library was created using the SMRTbell TM Template kit (version 2.0, Pacific Biosciences, Menlo Park, CA, USA), and the process included end repair, A-tail addition, sequencing junction addition, purification, and PCR amplification. The library was examined, authorized, and sequenced using the PacBio Sequel II platform.

#### 2.4.3. Bioinformatics Analysis

The raw data acquired by sequencing utilizing the Illumina sequencing technology contain a certain amount of low-quality data. To ensure the accuracy and reliability of the results of the subsequent analyses, reads containing low-quality bases (quality value ≤ 38) exceeding a certain percentage (the default is set to 40 bp), reads with a certain percentage of N bases (the default is set to 10 bp), and reads with an overlap with the adapter exceeding a certain threshold (set to 15 bp by default) are removed. If there is host contamination in the sample, it must be compared with the host sequence to filter out reads that might have come from the host [22,23,24]. (Bowtie2 software (version bowtie2 2.3.5.1) is used by default with the following parameters: –end-to-end, –sensitive, -I 200, -X 400).

PacBio’s raw sequencing reads are dumbbell-shaped sequences with junctions at both ends, also known as polymerase reads. The raw data are separated from the junctions and filtered out of the junction sequences to produce subreads, which are then filtered using the standard filtering subreads algorithm with a minimum length of 50. The ccs program (SMRT Link v8.0.0) (https://github.com/PacificBiosciences/ccs accessed on 5 December 2024) was used on the subreads to create high-precision HiFi reads [25], with the criterion of min-passes = 3, min-rq = 0.99, i.e., all readings had quality values of 0.99 and 0.99. Specifically, all readings have quality levels over Q20. In general, this section of the data constitutes appropriate input data for further analysis.

Following quality control, the clean data of each sample were utilized to perform genome assembly [26] using hifiasm-meta Flye (Version 2.8.1) [27], sequence files reflecting the samples’ basic genome were produced, and the assembly results were reviewed. Contigs ≥ 1 M in length were de-redundant using dRep (version 3.2.2) [28], and genome integrity was assessed using CheckM software (version 3.2.2) (https://ecogenomics.github.io/CheckM/ accessed on 5 December 2024) [29].

GeneMarkS software (version 4.17) [30] (http://topaz.gatech.edu/ accessed on 5 December 2024) predicted and filtered ORFs (open reading frames) based on the samples’ final assembly findings. The Repeat Masker program (version 4.0.5) [31] predicted scattered repeat sequences, whereas TRF (Tandem Repeats Finder, version 4.07b) [32] looked for tandem repeats in DNA sequences. The tRNAscan-SE program [33] can predict tRNA regions as well as secondary structures. The rRNA libraries were compared to identify rRNAs close to the database (identity default ≥ 50%), and the rRNAmmer program was used to anticipate emergent and unannotated rRNAs.

Rfam database comparison annotation was initially conducted using Rfam software (version 14.0) [34], and final sRNAs were determined using the cmsearch tool (version 1.1rc4). Gene islands were predicted using the IslandPath-DIOMB software (version 22) [35], which detects di-nucleotide bias in sequences (phylogenetically biased) and mobility genes (e.g., transposases or integrases) in sequences to determine gene islands and potential horizontal gene transfers.

The transposonPSI program [36] was used for transposon prediction. It employs PSI-Blast (version 2.6.0) homology contrast to identify potential transposon sequences in genomic or protein sequences. CRISPR (clustered regularly interspaced short palindromic sequence repeats) is a gene that has been found in 40% of sequenced bacteria and 90% of archaea. Clustered regularly interspaced short palindromic sequence repeats (CRISPR), found in 40% of sequenced bacteria and 90% of sequenced archaea, contain a short repetitive sequence homologous to both phages and plasmids, and are part of the prokaryotic immune system that is resistant to phages by acting on foreign homologous DNA and affecting plasmid ligation. CRISPR prediction of the sample genome was carried out using the CRISPRdigger (version 2.0) tool [37].

ORF (open reading frame) prediction was performed using Metagene Mark using Scaftigs (≥500 bp) of each sample and mixed assemblies, and information with length less than 100 nt was taken away from the prediction results. Prediction settings were set to default. CD-HIT software (version 4.8.1) was used to de-redundantize the ORF prediction results of each sample and mixed assembly, and clustering was performed by default with identity 95%, coverage 90%, and the longest sequence was selected as the representative sequence; the parameters used were the following: -c 0.95, -G 0, -aS 0.9, -g 1, -d 0. Using Bowtie2, the clean data of each sample were compared to the initial gene catalogue, and the number of reads of genes on the comparison in each sample was calculated; the comparison parameters were as follows: –end-to-end, –sensitive, -I 200, -X 400; the genes supporting the number of reads ≤ 2 were filtered out from the gene catalogue (Unigenes); the abundance information of each gene in each sample was calculated as follows. The gene catalogue (Unigenes) for further analysis was produced by filtering out the genes that supported ≤ 2 reads in each sample. The abundance information of each gene in each sample was determined from the number of reads and the length of the genes, and the calculation formula was as follows.(1)$$G_{k}=\frac{r_{k}}{L_{k}}\cdot \frac{1}{\sum_{i=1}^{n}\frac{r_{i}}{L_{i}}}$$

Unigenes were blasted against the eggNOG, KEGG, and CAZy functional databases using the DIAMOND software (version 3.5) [38] (blastp, e-value ≤ 1 × 10^−5^), and for each sequence BLAST (version 2.2.25) result, the comparison with the highest score (default identity ≥ 40%, and coverage ≥ 40%) for functional annotation was obtained. The DIAMOND program [38] aligned Unigenes with bacteria, fungi, archaea, and viruses sequences from NCBI’s NR database (version: 2018.01) (blastp, e-value < 1 × 10^−5^). For each sequence comparison result, those with e-value ≤ minimal e-value × 10 were selected for additional analysis. After filtering, many distinct species’ taxonomic information was collected, as one sequence may have numerous comparison findings. To ensure its biological significance, the LCA (Lowest Common Ancestor) algorithm [39] was used to take the taxonomic level prior to the occurrence of the first branch as the sequence’s species annotation information; from the LCA annotation results and gene abundance table, the abundance information of each sample at each taxonomic level (kingdom, phylum, class, order, family, genus, species) was obtained.

### 2.5. Ruminal Epithelial RNA Extraction and qRT-PCR

Rumen epithelium samples (100 mg) were collected and thoroughly pulverized with a low-temperature grinder (KZ-III-FP, Servicebio, Wuhan, China); total RNA of the rumen epithelium was extracted using the Trizol technique; and RNA concentration and purity were determined using an ultra-micro spectrophotometer (Nanodrop 2000, Thermo, Waltham, MA, USA).

Reverse transcription reaction: In a 20 μL reaction system, use 5 ¼ L of reaction buffer and 100 μM of Oligo (dT)18 primer, 0.5 μL of random hexamer primer (100 μM), 0.5 μL Servicebio^®^ RT Enzyme Mix, 1 μL total RNA, and 10 μL RNase-free water. Add to the 20 μL reaction system 10 μL RNase-free water. Add 20 μL, gently mix, and centrifuge. Reverse transcription PCR reaction protocol: 25 °C for 5 min, 42 °C for 30 min, and 85 °C for 5 s. Beijing Novogene Bioinformatics Technology Co., Ltd. (Beijing, China) produced the primers. The primer sequences are displayed in Table 2.

Take a 0.2 mL PCR tube and construct the following reaction system: 2X qPCR Mix 7.5 μL, 2.5 μM gene primer (upstream and downstream), 1.5 μL of reverse transcription product (cDNA), and 2.0 μL and 4.0 μL of water nuclease-free solution. The PCR reaction schedule is as follows: 95 °C for 30 s, 95 °C for 15 s, 60 °C for 30 s, and 40 cycles of 65 °C to 95 °C (0.5 °C per second).

### 2.6. Statistical Analysis

Changes in rumen short-chain fatty acids (SCFAs), microbial diversity, and relative abundance (across phylum, family, and genus levels) were analyzed using the independent samples t-test (significance threshold *p* < 0.05).

## 3. Results

### 3.1. Analysis of Rumen pH and Short-Chain Fatty Acids Content

Table 3 shows that Tarim wapiti rumen had significantly greater amounts of acetate content (61.77 mmol/L) and TVFA (77.32 mmol/L) than Karakul sheep, but pH (5.86) and propionate content (11.26 mmol/L) were significantly lower (*p* > 0.05).

### 3.2. Macrogenomic Analysis of Rumen Microbial Communities

#### 3.2.1. Analysis of the Rumen Structure of Microbial Communities

The PCoA (Figure 1a) and NMDS analysis (Figure 1b) showed that the rumen microbial community structure differed significantly between Tarim horse deer and Karakul sheep. The results of ANOISIM (Figure 1c) revealed that the *R*-value between groups of Tarim wapiti and Karakul sheep was 0.159, indicating that the difference between groups was greater than that within groups, and that there was a difference between rumen microorganisms of Tarim wapiti and Karakul sheep, but the difference was not significant (*p* > 0.05).

In this study, *Bacteroidetes* and *Firmicutes* were the dominant phyla in the portal microorganisms of Tarim wapiti and Karakul sheep, as indicated in Table 4. Tarim wapiti had a higher relative abundance of *Bacteroidetes* (46.13%) in their rumen microorganisms compared to Karakul sheep (34.77%) (*p* < 0.05). Karakul sheep had a higher relative abundance of *Firmicutes* (21.36%). *Proteobacteria* (8.23%) and *Actinobacteria* (0.51%) had significantly higher relative abundances in their rumen microorganisms (*p* < 0.05) compared to Tarim wapiti (Figure 2a).

As shown in Table 5, *Prevotellaceae* was the absolute dominant bacteria in the rumen of Tarim wapiti and Karakul sheep, with 21.45 and 20.06%, respectively, followed by *Bacteroidaceae* and *Lachnospiraceae*, with *Bacteroidaceae* being more dominant in the rumen microorganisms of Tarim wapiti (4.51%) than Karakul sheep (3.45%), and *Lachnospiraceae* on the contrary, with 1.84 and 3.60%, respectively. *Succinivibrionaceae* were more abundant in rumen microorganisms of Karakul sheep (4.41%) compared to Tarim wapiti (0.10%) (*p* < 0.05) (Figure 2b).

Table 6 shows that at the genus level, *Prevotella* and *Bacteroides* were both dominant bacteria in the rumen microorganisms of Tarim wapiti and Karakul sheep, with *Prevotella* being absolutely dominant, accounting for 17.62% and 16.74%, and *Bacteroides* 4.38% and 3.33%, respectively. Furthermore, Karakul sheep have a reasonably high abundance of *Methanobrevibacter* (1.52%) and *Succinivibrio* (1.47%) in their rumen microbes, with *Succinivibrio* being much more abundant than in Tarim wapiti (*p* < 0.05) (Figure 2c).

According to Table 6 and Table 7, the absolute dominant bacterium among the rumen microorganisms of Tarim wapiti was *Prevotella* sp. *ne3005* (2.62%), while other abundances accounting for more than 1% were *Prevotella ruminicola* (2.41%), *Bacteroidales bacterium WCE2004* (1.65%), *Bacteroidales bacterium WCE2008* (1.50%), *Prevotella* sp. *tc2-28* (1.26%), *Bacterium F083* (1.25%), *Prevotella* sp. *tf2-5* (1.19%), and *Selenomonas ruminantium* (1.10%). *Prevotella* sp. *ne3005* was the most common rumen microbe in Karakul sheep (2.65%), followed by *Prevotella ruminicola* (2.82%), *Prevotella* sp. *tc2-28* (1.95%), *Selenomonas ruminantium* (1.53%), *Succinivibrio dextrinosolvens* (1.47%), and *Succiniclasticum ruminis* (1.02%). In Tarim wapiti, *Bacterium F083* was much more abundant in the rumen microorganisms than in Karakul sheep, whereas *Succinivibrio dextrinosolvens* was significantly less abundant than in Karakul sheep (*p* < 0.05) (Figure 2d).

The LEfSe research found substantial variations in biomarkers between rumen microbes of Tarim wapiti and Karakul sheep. Tarim wapiti and Karakul sheep had statistically distinct biomarkers, which were classified into two phyla, four orders, three orders, five families, nine genera, and thirteen species (Figure 3). Significantly different biomarkers were found in Tarim wapiti rumen microorganisms, including *Bacteroidetes*, unclassified *Bacteria*, *Cytophagia*, *Muribaculaceae*, *Lentimicrobiaceae*, unclassified *Methanomicrobiales*, and related orders, phyla, families, and genera belonging to 23 taxa. *Gammaproteobacteria*, *Erysipelotrichia*, *Succinivibrionaceae*, *Ruminobacter*, *Bifidobacterium*, and *Bifidobacterium*, and related orders, phyla, families, and genera belonging to 13 taxa in Karakul sheep’s rumen, were also found.

#### 3.2.2. Annotated Analysis of Rumen Microbial Functions

Functional annotation through KEGG functional data indicated that the metabolic pathway with the largest number of annotated genes at the first level (level 1) (Figure 4a) was metabolism in both Tarim wapiti and Karakul sheep, with 11.88% and 11.17%, respectively. At level 2, metabolism accounted for 60% of the top ten metabolic pathways in terms of relative abundance, with carbohydrate metabolism being the highest (Tarim wapiti: 3.45%; Karakul sheep: 3.24%), followed by amino acid metabolism (Tarim wapiti: 2.68%; Karakul sheep: 2.51%), then metabolism of cofactors and vitamins (Tarim wapiti: 1.99%; Karakul sheep: 1.99%), energy metabolism (Tarim wapiti: 1.90%; Karakul sheep: 1.73%), nucleotide metabolism (Tarim wapiti: 1.87%; Karakul sheep: 1.69%), and glycan biosynthesis and metabolism (Tarim wapiti: 1.10%; Karakul sheep: 0.92%). Among them, Tarim wapiti had considerably higher levels of rumen bacteria, including glycan biosynthesis and metabolism and replication and repair, compared to Karakul sheep (*p* < 0.05) (Figure 4b).

In CAZy functional annotation, the abundance of glycoside hydrolases (GHs) and carbohydrate esterases (CEs) in the rumen of Tarim wapiti was higher than that in Karakul sheep, whereas the abundance of glycosyl transferases (GTs), carbohydrate-binding modules (CBMs), polysaccharide lyases (PLs), and auxiliary activities (AAs) was lower than that in Karakul sheep (Figure 5a). LDA (Figure 5b) revealed substantial differences between Tarim wapiti and Karakul sheep in GHs, GTs, and CEs, with Tarim wapiti considerably enriched in GH20, GH33, GH92, GH97, and GT3, and Karakul sheep significantly enriched in GH10, GH115, GT2, and CE4.

### 3.3. Differential mRNA Expression of Transporters and pH-Regulation-Related Proteins Involved in SCFA Absorption in the Rumen Epithelium

The relevant carriers involved in SCFA uptake are the following: PAT1, DRA, AE2, MCT1, and MCT2; the relevant proteins involved in pH regulation are the following: Na^+^/K^+^ ATPase, NHE3, and NHE4. Each gene’s amplification curve was smooth and S-shaped, which met the criteria for additional data analysis Figure 6.

The 2-ΔΔCT method was used to calculate the relative mRNA expression of each vector and protein, with GAPDH as the internal reference. The Tarim wapiti’s rumen epithelium showed significantly higher expression of the relevant vectors *PAT1*, *DRA*, *AE2*, and *MCT2*, as well as the relevant proteins *Na^+^/K^+^ ATPase* involved in pH regulation, compared to the Karakul sheep (*p* < 0.05 or *p* < 0.01) (Figure 7).

When combined with rumen bacteria and short-chain fatty acids, the Tarim wapiti demonstrated greater rumen microorganisms–short-chain fatty acids–epithelial genes interactions (Figure 8).

## 4. Discussion

### 4.1. Analysis of Rumen pH and Short-Chain Fatty Acids Content

Microbial fermentation in the rumen degrades and converts nutrients that cannot be directly or readily consumed into tiny molecules such as short-chain fatty acids, NH_3_-N, and MCP [40,41]. Ruminal pH is maintained by synergistic acid-base regulation by the rumen microbial population and host rumen metabolism [42], and the level of rumen pH influences rumen microbial activity, feed digestibility, and rumen health. Generally, the rumen pH ranges from 6.0 to 7.0 and is slightly acidic. In the present study, the rumen pH in Tarim wapiti and Karakul sheep was 5.86 and 6.25, respectively, with Tarim wapiti having a considerably lower pH than Karakul sheep. Saliva, ruminal short-chain fatty acids content, and ruminal epithelial absorption all have an impact on ruminal pH. Furthermore, a negative connection was found between rumen pH and volatile fatty acid content [43]. In the current study, the rumen TVFA content in Tarim wapiti was substantially greater than that in Karakul sheep, resulting in the long-term preservation of low pH in Tarim wapiti rumens. It has also been demonstrated that rumen pH below 6.0 reduces the activity of crude fiber-degrading bacteria, resulting in poor roughage digestibility [17]. Numerous studies have revealed that the rumen pH of Tarim wapiti has been below 6.0 for a long period [2,7,8,44]. In the current study, the rumen pH of Tarim wapiti was 5.86, which was likewise less than 6.0. Although the Tarim wapiti’s rumen pH was low for a long time, it nevertheless maintained a high feed utilization efficiency, which might be attributed to the rumen’s adaptability to feed on alkaline-tolerant plants for a long length of time.

### 4.2. Analysis of the Rumen Structure of Microbial Communities

Rumen microorganisms may ferment items that cannot be consumed directly to deliver nutrients and energy to the host. The host’s genetics are the most important genetic component controlling the microbiota [44]. Furthermore, it has been established that species have distinct dominating floras under the same food and husbandry management settings [3,14,45]. In the current study, Bacteroidetes, Prevotellaceae, and Prevotella were the dominant rumen bacteria in both Tarim wapiti and Karakul sheep rumens, but their abundance was higher in the Tarim wapiti rumen, which was consistent with the findings of Qian [3] and Glendinning [14], who studied the differences in rumen microorganisms between different species. Furthermore, the dominating bacteria in the Tarim wapiti’s rumen were structurally comparable to earlier research on Tarim wapiti [2,3,7,8], Sika deer [46], elk [47], and roe deer [48]. This further shows that genetic and metabolic variables impact the organization and diversity of host rumen microorganisms [49].

*Bacteroidetes* have been found to ferment polysaccharides from the host’s indigestible fiber cell walls and supply energy to the host [50,51], with approximately 20% of the genome devoted to the transport and catabolism of a wide range of polysaccharides and the regulation of these processes [52], and Prevotella, the highest abundance of *Bacteroidetes*, is capable of degrading hemicellulose, pectin, starch, and proteins to produce succinate and acetate as host energy [53,54].

*Bacteroides* were discovered to be abundant in the liquid-phase rumen contents of Tarim wapiti [3,7,8] and Canadian cervids [55], and Bacteroidetes accounted for a larger percentage of the rumen in animals given roughage [56]. Furthermore, by comparing the process of diet-induced microbial adaptation in red deer during different introduction periods, it was discovered that wild release significantly altered the gut microbial structure of horse deer, with *Bacteroidetes* serving as a key strain for adaptation to complex food resources [57]. *Prevotella* of *Bacteroidetes* was discovered to have an important role in the rumen fermentation process of sika deer [46] and in the high utilization efficiency of low-quality roughage by Tarim wapiti [3]. *Bacteroidetes*, particularly *Prevotella*, appear to be the most important bacteria influencing host nutrition consumption and adaptability to environmental food supplies. In this study, the abundance of *Bacteroidetes* in the rumen of Tarim wapiti was significantly higher than that in Karakul sheep, and the abundance of *Prevotella* was also higher than that in Karakul sheep, but the abundance of *Succinivibrion*, *Succiniclasticum*, and *Succinivibrio dextrinosolvens* in the rumen of Karakul sheep was significantly higher than that in the Tarim wapiti. *Succinivibrion* was discovered to ferment succinic acid and convert it into propionate [58]. Starch and soluble carbohydrates generate more propionate, whereas fiber and hemi-fibrous materials generate more acetate [17]. Thus, the Tarim wapiti’s superior efficiency of digestive use of low-quality roughage may be ascribed to the larger abundance of *Bacteroidetes* and *Prevotella* in the rumen, whereas Karakul sheep chose to use starch and soluble sugars for energy supply to the host.

In addition, Proteobacteria are important markers of the balance of gut flora in animals [59]. *Actinobacteriota* can produce metabolic chemicals that serve a variety of biological functions, including the preservation of the internal environment of the intestinal barrier, antimicrobials, antivirals, and immune augmentation [60,61,62]. *Bifidobacterium*, for example, plays a vital role in regulating host intestinal and immunological homeostasis [63]. *Eggerthellaceae* and *Nocardiaceae* were discovered to be antibiotic-producing biomarkers in *Actinobacteriota* [64]. It has been discovered that an excessive amount of roughage causes disarray of host gut microorganisms [65,66]. In the current study, *Proteobacteria* and *Actinobacteriota* were considerably abundant in the rumen of Karakul sheep, whereas *Gammaproteobacteria* and *Bifidobacterium* were found as biomarkers when compared to Tarim wapiti. It is notable that the Tarim wapiti’s intestinal rumen microbial homeostasis outperformed that of the Karakul sheep and stayed steady when given high fiber levels. Furthermore, we discovered that Tarim wapiti rumen biomarkers belong to the order, family, and genus unclassified *Bacteria*, *Muribaculaceae*, and *Lentimicrobiaceae*. Similarly, an analysis of the fecal microorganisms of deer, yak, sheep, and camels revealed that *Lentimicrobiaceae* were abundant in deer feces [67]. *Muribaculaceae* were discovered to have numerous applications in complicated carbohydrate breakdown [68]. As a result, *Muribaculaceae* and *Lentimicrobiaceae* may be taxa influencing the digestibility of low-quality roughage in Tarim wapiti, and unclassified *Bacteria* may potentially hold tremendous promise for further investigation.

### 4.3. Annotated Analysis of Rumen Microbial Functions

Rumen microbial communities degrade indigestible lignin and cellulose to produce absorbable nutrients and critical metabolites. Despite the great degree of similarity in rumen microbial makeup between species, there may be distinct roles that allow various species to digest different lignin and cellulose [69]. We discovered differences in metabolic pathways and CAZymes between Tarim wapiti and Karakul sheep using functional annotation. Glendinning et al. [14] discovered that while most CAZymes and KEGG immediate homologs were present in all ruminant species, there was a significant difference in abundance of CAZymes and KEGG immediate homologs among species, which was consistent with the findings of this study. The number of KEGG orthologs varied greatly between species, consistent with the findings of our study.

CAZymes perform complicated activities including glycosidic bond breakdown, modification, and creation, all of which are critical for host nutrition acquisition. GHs may hydrolyze glycosidic linkages in complex carbohydrates, degrade cellulose, hemicellulose, and starch effectively [70], and degrade lignocellulosic materials synergistically with CEs and PLs [71,72]. Furthermore, it was discovered that cows with high feed consumption efficiency had a greater abundance of GHs, whereas ineffective rumens had more GTs [73]. In this study, the Tarim wapiti rumen also had a higher abundance of GHs, while the Karakul sheep rumen had more GTs; LDA discriminant analysis revealed that GH20, GH33, GH92, GH97, and GT3 were significantly enriched in the Tarim wapiti rumen, whereas GH10, GH115, GT2, and CE4 were significantly enriched in the Karakul sheep rumen. It was discovered that there were considerable changes in CAZymes abundance between species, with GH97 and GH92 being highly enriched in the rumen GHs of red deer and reindeer [14]. Bacteroides was discovered to be engaged in GH92, which encodes mannosidases that play a role in the degradation of N-glycans [74], and a substantial fraction of CAZymes belong to the Bacteroidaceae (40%) [75]. Prevotella was also a substantial donor of GHs in the bovine rumen [76]. It is worth noting that in this study, the rumen of Tarim wapiti had a higher abundance of Bacteroidaceae and Prevotella than that in Karakul sheep; thus, we identified the key role of Bacteroidetes and Prevotella among them in the rumen of Tarim wapiti to contribute more GHs for improving feed digestion and utilization.

### 4.4. Differential mRNA Expression of Transporters and pH-Regulation-Related Proteins Involved in SCFA Absorption in the Rumen Epithelium

The content of rumen short-chain fatty acids is intimately connected to fermentation by rumen bacteria and epithelial absorption. The MCT family of SCFA^−^ and HCO_3_^−^ exchange carriers (DRA, PAT1, and AE2) have been identified as the carriers implicated in SCFA uptake in the rumen epithelium [77]. Furthermore, 50% of SCFA in the rumen is absorbed by SCFA^−^–HCO_3_^−^ exchange [78,79], particularly acetate [80] and butyrate [81]. Furthermore, rumen epithelial cell homeostatic regulatory proteins (*NHE* family, *Na^+^/K^+^ ATPase*, etc.) regulate intracellular pH, which aids in SCFA absorption.

The rumen epithelium of sika deer was reported to express *MCT1* and *MCT4* [82]. In this study, we discovered that the related vectors *PAT1*, *DRA*, *AE2*, and *MCT2* involved in SCFA absorption, as well as the related protein *Na^+^/K^+^ ATPase* involved in pH regulation, were expressed in the rumen epithelium of Tarim wapiti, and the relative expressions were also significantly higher than that in Karakul sheep, with no significant differences in relative expressions of *MCT1*, *NHE3*, and *NHE4*. This indicated that the Tarim wapiti has a higher absorption capacity for short-chain fatty acids. It was discovered that dietary fiber levels influenced the amount of short-chain fatty acids in the rumen, which in turn changed the expression levels of *AE2*, *MCT1*, and *Na^+^/K^+^ ATPase* [82]. In the current investigation, the feeding levels of both animals were identical. Theoretically, the Tarim wapiti’s rumen epithelial short-chain fatty acids absorption capacity was greater than that of Karakul sheep, and the rumen short-chain fatty acids content should have been lower, but the rumen short-chain fatty acids content of Tarim wapiti was higher than that of Karakul sheep. As a result, it is claimed that Tarim wapiti rumen bacteria fermented the diet more than Karakul sheep. As a result, we integrated rumen microorganisms, short-chain fatty acids, and rumen epithelial short-chain fatty acids uptake and transport-related vectors to reveal that the Tarim wapiti has greater rumen microbe–short-chain fatty acids–epithelial linkages.

## 5. Conclusions

The Tarim wapiti’s rumen contained more contents of acetate and TVFA compared to the Karakul sheep, whereas the ruminal pH and propionate content were opposite. *Bacteroidetes*, *Prevotellaceae*, and *Prevotella* were the most abundant bacteria in the rumens of Tarim wapiti and Karakul sheep. *Bacteroidetes* was significantly enriched, and the relative abundance of glycan biosynthesis and metabolism as well as replication and repair was higher in the rumen of Tarim wapiti, as well as having higher abundance of GHs, with significant abundance of GH20, GH33, GH92, GH97, and GT3. Meanwhile, Tarim wapiti had considerably more expression of *PAT1*, *DRA*, *AE2*, and *MCT2*, vectors implicated in SCFA absorption, and *Na^+^/K^+^ ATPase*, proteins involved in pH control, in the rumen epithelium compared to Karakul sheep.

Overall, the combination of rumen bacteria, VFA content, and rumen epithelial VFA uptake-related vectors demonstrated that Tarim wapiti had more interactions with rumen microbes, short-chain fatty acids, and the rumen epithelium. These might be critical components in the high-efficiency short-chain fatty acids of roughage digestion and utilization. These findings may be used as a theoretical foundation for Tarim wapiti to use high roughage and ration formulation more efficiently.

## Figures and Tables

**Figure 1 microorganisms-13-01111-f001:**
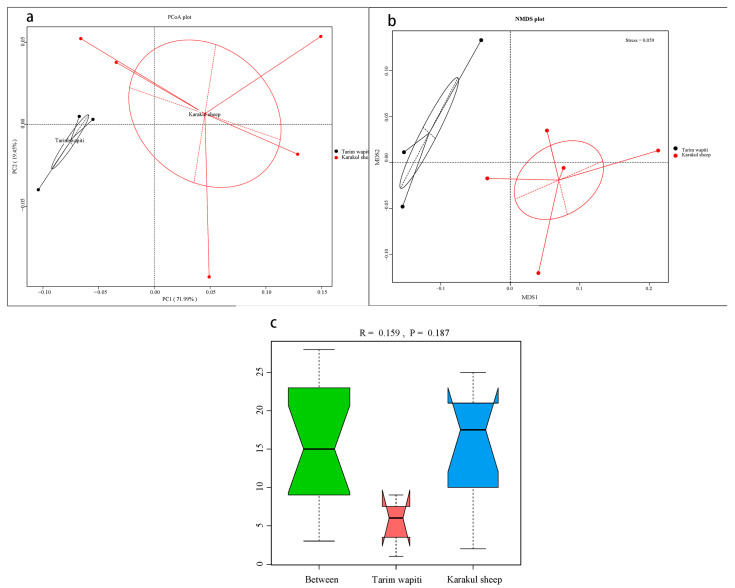
Beta diversity analysis of rumen microorganisms in Tarim wapiti and Karakul sheep; (**a**) PCoA (principal coordinate analysis) analysis; (**b**) NMDS (non-metric multidimensional scaling) analysis; (**c**) ANOSIM analysis.

**Figure 2 microorganisms-13-01111-f002:**
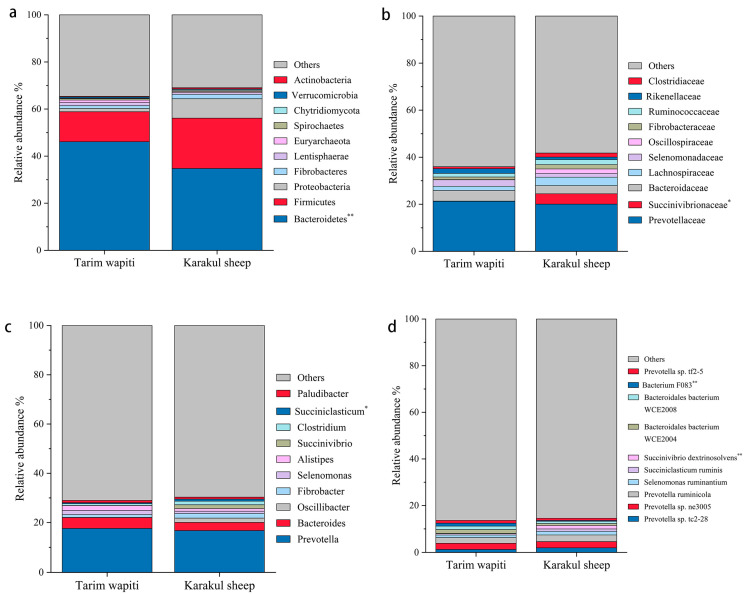
The relative abundance of microbial in the rumen of Tarim Wapiti and Karakul sheep. (**a**) phyla level. (**b**) family level. (**c**) gene level. (**d**) bacterium level. Note: ** indicate significant differences (*p* < 0.01), * indicate significant differences (*p* < 0.05).

**Figure 3 microorganisms-13-01111-f003:**
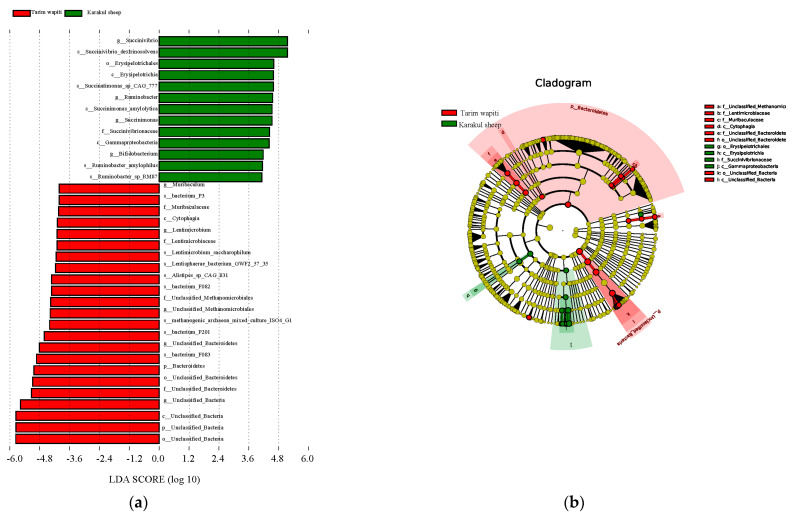
LEfSe study of significant biomarker differences between Tarim wapiti and Karakul sheep; (**a**) LDA value > 3. (**b**) The classification level of the circle from the outside to the inside is in order of phyla, class, order, family, genus, and species. Different color points in the phylogenetic tree stand for bacteria which are important in each group, respectively. Yellow points indicate taxa, which have no significant differences among groups.

**Figure 4 microorganisms-13-01111-f004:**
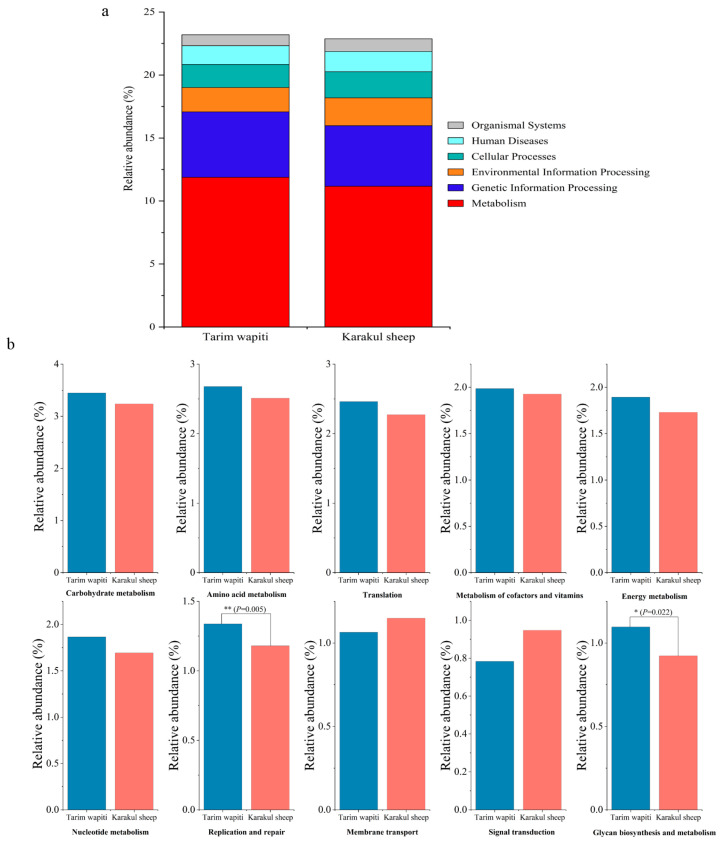
Annotated changes in KEGG functions of rumen microbes between Tarim wapiti and Karakul sheep; (**a**) KEGG pathway level 1; (**b**) KEGG pathway level 2.

**Figure 5 microorganisms-13-01111-f005:**
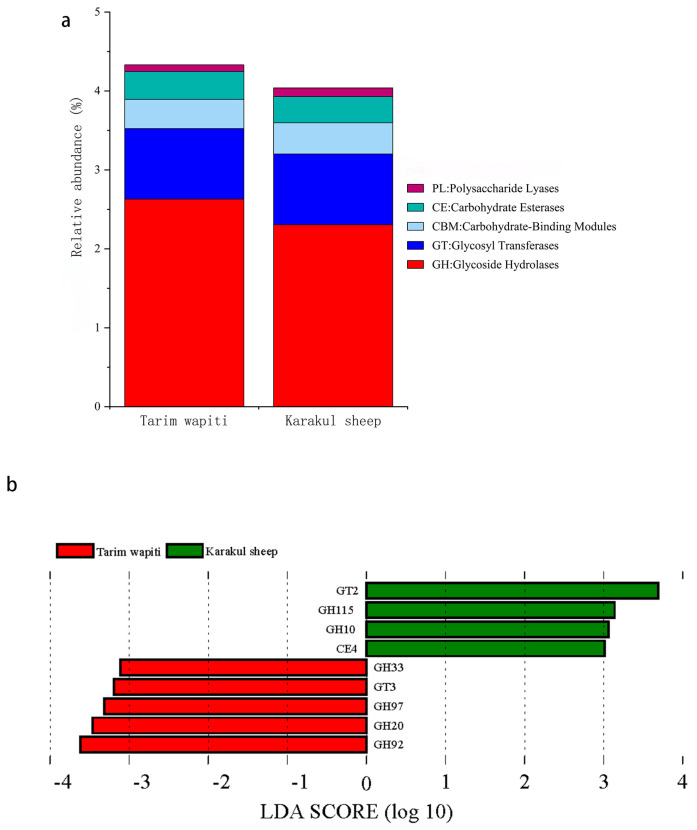
Comparison of CAZyase functional annotations in rumen microorganisms from Tarim wapiti and Karakul sheep. (**a**) Functional annotation of CAZyase at level 1; (**b**) distribution of LDA values for CAZyase differences (LDA score > 3).

**Figure 6 microorganisms-13-01111-f006:**
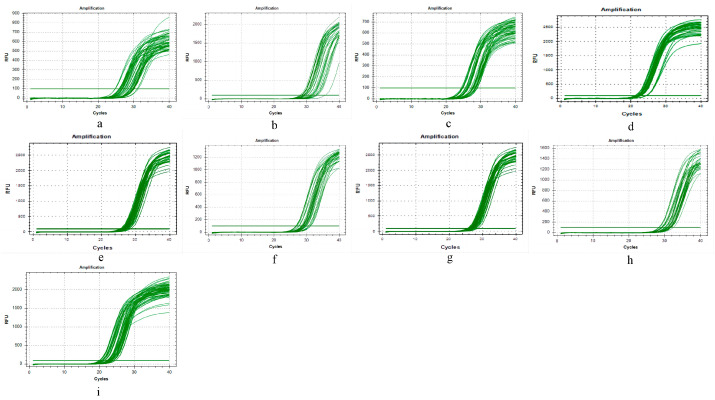
qRT-PCR amplification curves for related vectors and proteins. (**a**) *PAT1*; (**b**) *DRA*; (**c**) *AE2*; (**d**) *MCT1*; (**e**) *MCT2*; (**f**) *Na^+^/K^+^ ATPase*; (**g**) *NHE3*; (**h**) *NHE4*; (**i**) *GAPDH*.

**Figure 7 microorganisms-13-01111-f007:**
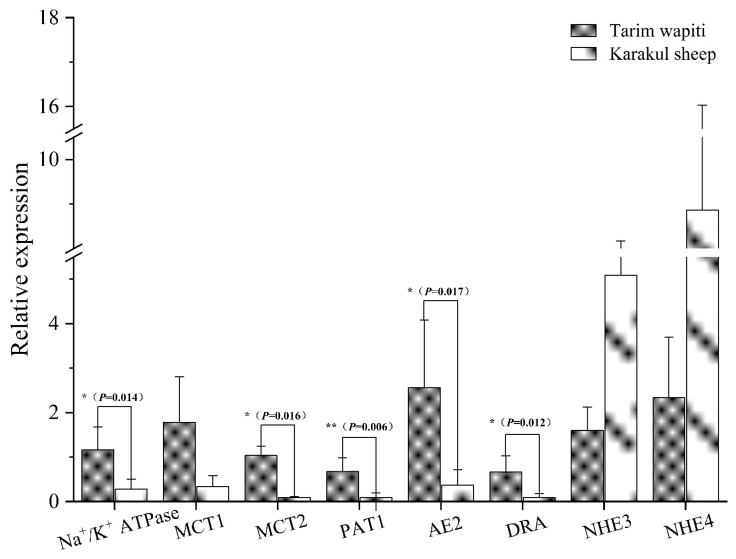
The rumen epithelium is involved in the mRNA expression of SCFA-related vectors and pH-regulation-related proteins. Note: ** indicate significant differences (*p* < 0.01), * indicate significant differences (*p* < 0.05).

**Figure 8 microorganisms-13-01111-f008:**
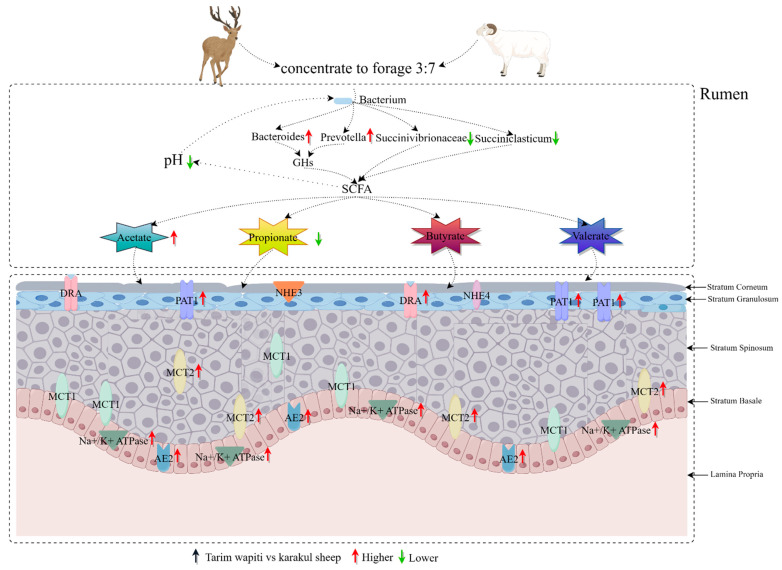
Interactions between rumen microorganisms, short-chain fatty acids, and epithelial genes.

**Table 1 microorganisms-13-01111-t001:** Composition and nutritional level of diets.

Items	Value
Composition	Content (%)
forage	
Corn stalks	19.5
Wheat stalks	13
Alfalfa	22.5
Cottonseed hull	15
concentrate	
Corn	23.5
Soybean meal	2.5
Cotton meal	2.5
Salt	0.5
Premix ^¶^	1
Total	100
**Nutrient Levels**	
Digestible energy (DE) ^§^, MJ/kg	11.1
Crude protein (CP)%	11.51
Neutral detergent fiber (NDF), %	53.7
Acid detergent fiber (ADF), %	33.8
Calcium (Ca)%	0.53
Phosphorus (P)%	0.24

^¶^ Premix: VA 940 IU/kg, VE 20 IU/kg, minerals (mg/kg): S 200 mg/kg, Fe 24 mg/kg, Cu 8 mg/kg, Mn 40 mg/kg, Zn 40 mg/kg, I 0.3 mg/kg, Se 0.2 mg/kg, Co 0.1 mg/kg. ^§^ DE: The digestive energy was calculated using the digestive energy of sheep in the *Animal Nutritional Parameters and Feeding Standards*.

**Table 2 microorganisms-13-01111-t002:** Primer sequences.

Genes	Sequences (5′–3′)	Amplicon Size/bp	Annealing Temperature/°C
*HMGCS1*	F:CGAGCACTACAGCCGAGCATA R:CCTGAAGTCCTCCACCTCACAG	136	60
*HMGCS2*	F:CAGGCTGCTGTGTCCAATGC R:GTACCTGGAGCGAGTGGATGAG	172	
*SLC27A6*	F:TCACGCTCTAATTGCTCATCCG R:CGGTTCTGCCATTGCTCTCC	207	
*SLC26A9*	F:CCTCGCTCATCTTCGCTCTCAT R:GCTGTTGCCACCACTACCACAA	127	
*NHE3*	F:AGAAGGTCAGCTCAGAAGTCTCG R:GCTGGGAGAACGGATGAAAG	121	
*NHE4*	F:CGACATTTTGGCTGGATGTG R:CTGGGTGAAACGGGTGATAAA	106	
*Na*^+^/*K*^+^-*ATPase*	F:ATCACGGGTGTGGCTGTGT R:TGATGCCGATGAGGAAGATG	122	
*MCT1*	F:CATACCAGGGGTTTATTGATGGA R:GTAACGGAACACTGAAAATGGATG	257	
*MCT2*	F:TGTTATGCTGTTTGGGTATGGTCT R:TGTTAAGGCAGGTTGCAGGTT	119	
*PAT1*	F:CTGGTGAAGCTCCTGAATGAAA R:CACGATGTCCACCCCAAA	135	
*AE2*	F:TCAACGCCTTCCTGGACTG R:CCTGCTCCTCCCCTCTTTCT	120	
*GAPDH*	F:ACCACTGTCCACGCCATCAC R:ACGCCTGCTTCACCACCTTC	271	

**Table 3 microorganisms-13-01111-t003:** Comparing the rumen pH and VFA content in Tarim wapiti and Karakul sheep.

Items	Groups		SEM	*p*-Value
	Tarim Wapiti	Karakul Sheep		
pH	5.86	6.25 **	0.09	0.006
Acetate, mmol/L	61.77 **	49.93	1.92	0.001
Propionate, mmol/L	11.26	13.85 **	0.5	0.002
Butyrate, mmol/L	2.81	3.24	0.36	0.29
Valerate, mmol/L	1.49	1.2	0.35	0.456
TVFA, mmol/L	77.32 **	68.23	1.77	0.002

Note: ** indicate significant differences (*p* < 0.01).

**Table 4 microorganisms-13-01111-t004:** The relative abundance of microbial phyla in the rumen of Tarim Wapiti and Karakul sheep.

Phyla	Groups		SEM	*p*-Value
	Tarim Wapiti	Karakul Sheep		
*Bacteroidetes*	46.13 **	34.77	0.98	<0.01
*Firmicutes*	12.73	21.36	5.42	0.25
*Proteobacteria*	1.40	8.23 *	0.54	0.048
*Fibrobacteres*	1.16	1.87	0.95	0.53
*Lentisphaerae*	1.29	0.66	0.22	0.10
*Euryarchaeota*	1.11	0.46	0.16	0.06
*Spirochaetes*	0.61	0.45	0.22	0.54
*Chytridiomycota*	0.18	0.47	0.24	0.35
*Verrucomicrobia*	0.51	0.29	0.01	0.15
*Actinobacteria*	0.26	0.51 *	0.04	0.039

Note: ** indicate significant differences (*p* < 0.01), * indicate significant differences (*p* < 0.05).

**Table 5 microorganisms-13-01111-t005:** The relative abundance of microbial families in the various rumen of Tarim wapiti and Karakul sheep.

Families	Groups		SEM	*p*-Value
	Tarim Wapiti	Karakul Sheep		
*Prevotellaceae*	21.45	20.06	1.58	0.50
*Succinivibrionaceae*	0.10	4.41 *	0.30	0.04
*Bacteroidaceae*	4.51	3.45	0.24	0.13
*Lachnospiraceae*	1.84	3.60	0.92	0.30
*Selenomonadaceae*	2.75	1.60	0.98	0.41
*Oscillospiraceae*	0.08	1.88	1.35	0.41
*Fibrobacteraceae*	1.15	1.86	0.94	0.56
*Ruminococcaceae*	1.54	2.12	0.47	0.35
*Rikenellaceae*	2.02	1.08	0.23	0.15
*Clostridiaceae*	0.85	1.71	0.30	0.15

Note: * indicate significant differences (*p* < 0.05).

**Table 6 microorganisms-13-01111-t006:** Analysis of rumen microbes at the genus level in Tarim wapiti and Karakul sheep.

Genera	Groups		SEM	*p*-Value
	Tarim Wapiti	Karakul Sheep		
*Prevotella*	17.62	16.74	0.56	0.40
*Bacteroides*	4.38	3.33	0.51	0.37
*Oscillibacter*	0.07	1.74	1.59	0.42
*Fibrobacter*	1.15	1.86	0.98	0.18
*Selenomonas*	1.68	0.91	0.88	0.73
*Alistipes*	1.96	1.06	1.11	0.89
*Succinivibrio*	0.06	1.57	0.39	0.18
*Clostridium*	0.71	1.46	0.61	0.29
*Succiniclasticum*	0.43	0.85 *	0.09	0.02
*Paludibacter*	0.93	0.81	0.72	0.67

Note: * indicate significant differences (*p* < 0.05).

**Table 7 microorganisms-13-01111-t007:** Analysis of the rumen microbial species level microbiota structure in Tarim wapiti and Karakul sheep.

Species	Groups		SEM	*p*-Value
	Tarim Wapiti	Karakul Sheep		
*Prevotella* sp. *tc2-28*	1.26	1.95	1.11	0.56
*Prevotella* sp. *ne3005*	2.62	2.65	0.85	0.97
*Prevotella ruminicola*	2.41	2.82	0.43	0.38
*Selenomonas ruminantium*	1.10	1.53	0.80	0.60
*Succiniclasticum ruminis*	0.56	1.02	0.50	0.39
*Succinivibrio dextrinosolvens*	0.10	1.47 **	0.25	<0.01
*Bacteroidales bacterium WCE2004*	1.65	0.85	0.47	0.14
*Bacteroidales bacterium WCE2008*	1.50	0.99	0.40	0.26
*Bacterium F083*	1.25 **	0.34	0.20	<0.01
*Prevotella* sp. *tf2-5*	1.19	0.93	0.29	0.40

Note: ** indicate significant differences (*p* < 0.01).

## Data Availability

The data that support the findings of this study are openly available in NCBI. https://www.ncbi.nlm.nih.gov/sra/PRJNA1110073 accessed on 5 December 2024.

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
