# Peer review of "Comparative Analysis of Microbial–Short-Chain Fatty Acids–Epithelial Transport Axis in the Rumen Ecosystem Between Tarim Wapiti (Cervus elaphus yarkandensis) and Karakul Sheep (Ovis aries)"

_microorganisms, 2025, doi:10.3390/microorganisms13051111_

Round 1
Reviewer 1 Report
Comments and Suggestions for Authors
- Abstract written correctly, you should write the purpose of the study
- Introduction written correctly, the purpose should be modified because it does not correspond to the study's objectives
- Material and methods of research: please justify why karakul sheep were chosen. On the basis of what literature data was selected ration (type of feed). At what age were the animals, in what conditions of maintenance ( in the sheepfold or in the pasture)
- Whether the study was approved by the ethics or bioethics committee
- Results described correctly, documented with diagrams and figures
- The discussion should add what practical significance the research has, whether it can be applied to all ruminants

Author Response
Dear Editors and Reviewers:
We appreciate your invitation to revise our manuscript again and the valuable comments by the reviewers. The manuscript has been carefully revised according to the comments.
Reviewer: 1
Comments 1: [Abstract written correctly, you should write the purpose of the study]
Response 1: [Thank you for pointing this out. We agree with this comment. Therefore, we have reiterated the purpose of the study in the abstract] Lines 19-23.
Comments 2: [Introduction written correctly, the purpose should be modified because it does not correspond to the study's objectives]
Response 2: [Thank you for pointing this out. We agree with this comment. Therefore, we have reiterated the purpose of the study in the abstract and introduction] Lines 19-23, 86-90.
Comments 3: [Material and methods of research: please justify why karakul sheep were chosen. On the basis of what literature data was selected ration (type of feed). At what age were the animals, in what conditions of maintenance ( in the sheepfold or in the pasture)]
Response 3: [Thank you for pointing this out. We agree with this comment. Therefore, We have indicated in the Material and methods of research the literature on which the diet (feed type), sheep age, and feeding circumstances are selected] Lines 93-103.
Comments 4: [Whether the study was approved by the ethics or bioethics committee]
Response 4: [Thank you for pointing this out. We agree with this comment. Therefore, we have included information on obtaining approval from an ethics or bioethics committee in the manuscript]. Lines 547-549.
Comments 5: [Results described correctly, documented with diagrams and figures]
Response 5: [Thank you for pointing this out. We agree with this comment]
Comments 6: [The discussion should add what practical significance the research has, whether it can be applied to all ruminants]
Response 6: [Thank you for pointing this out. We agree with this comment. Therefore, because of its peculiar environment, the Tarim wapiti has evolved distinct digestive physiology and energy metabolism. The current work focuses on understanding the mechanism of high roughage digestive efficiency in Tarim wapiti and if it is relevant to all ruminants, with additional in-depth research planned later]

Reviewer 2 Report
Comments and Suggestions for Authors The article is interesting and innovative, I suggest some minor changes Title - I believe that authors should not include acronyms in the title. - AGV should be changed to short-chain fatty acids; Note this throughout the text. Keywords: Remove the words contained in the title. The research title already serves as a search, so the keywords should be expressions that can contribute to more search results, in addition to the title. Introduction: -Insert information about the animals studied Tarim Wapiti (Cervus elaphus yarkandensis) and Karakul Sheep. Since the paper will be read by researchers from different locations, it is advisable to provide descriptive information about the animals. -Insert a hypothesis before the objective Methodology -How many animals (Tarim wapiti) were used? Include weight and age, as presented for the Karakul Sheep. -Why was this time chosen to feed the animals (7 p.m.) and not an afternoon time? What recommendations were used to prepare the diet? Lines 98-101 - Describe in more detail the analyses performed; -The nutritional composition of the diet ingredients should be included; -Include the levels of total carbohydrates, non-fibrous carbohydrates, neutral detergent fiber and acid detergent fiber; -Describe how the digestible energy was calculated _After how long in confinement were the animals slaughtered? How was the slaughter carried out? -Describe the locations where the rumen samples were collected; _Include the citation and reference for each procedure used in the methodologyAuthor Response
Dear Editors and Reviewers:
We appreciate your invitation to revise our manuscript again and the valuable comments by the reviewers. The manuscript has been carefully revised according to the comments.
Reviewer: 2
Comments 1: [The article is interesting and innovative, I suggest some minor changes Title - I believe that authors should not include acronyms in the title. - AGV should be changed to short-chain fatty acids; Note this throughout the text]
Response 1: [Thank you for pointing this out. We agree with this comment. Therefore, We have modified that VFAs should be changed to short-chain fatty acids in the title and body of the text]
Comments 2: [Keywords: Remove the words contained in the title. The research title already serves as a search, so the keywords should be expressions that can contribute to more search results, in addition to the title]
Response 2: [Thank you for pointing this out. We agree with this comment. Therefore, We have removed the words from the title and added phrases that can contribute to more search results] Line 40.
Comments 3: [Introduction: -Insert information about the animals studied Tarim Wapiti (Cervus elaphus yarkandensis) and Karakul Sheep]
Response 3: [Thank you for pointing this out. We agree with this comment. Therefore, We have insert information about the animals under study in the introduction, and Specific feeding and other specifics are provided in the materials and methods] Lines 42-58 .
Comments 4: [Since the paper will be read by researchers from different locations, it is advisable to provide descriptive information about the animals. -Insert a hypothesis before the objective Methodology -How many animals (Tarim wapiti) were used? Include weight and age, as presented for the Karakul Sheep. -Why was this time chosen to feed the animals (7 p.m.) and not an afternoon time? What recommendations were used to prepare the diet? Lines 98-101 - Describe in more detail the analyses performed; -The nutritional composition of the diet ingredients should be included; -Include the levels of total carbohydrates, non-fibrous carbohydrates, neutral detergent fiber and acid detergent fiber; -Describe how the digestible energy was calculated _After how long in confinement were the animals slaughtered? How was the slaughter carried out? -Describe the locations where the rumen samples were collected; _Include the citation and reference for each procedure used in the methodology]
Response 4: [hank you for pointing this out. We agree with this comment. Therefore, 1. We have labeled how many animals were used, including weight and age; feeding at 7 pm was chosen to take into account local feeding habits.
- We have described in greater detail the nutrient composition of the diet ingredients, including neutral detergent fiber and acid detergent fiber, as well as how digestible energy was calculated; however, because this study was a follow-up to a previous study that focused on the high digestive utilization efficiency of roughage and high-fiber forage by Tarim wapiti, total carbohydrates and non-fiber carbohydrates were not taken into account in the formulation.
- We have described how the animals were slain and where rumen samples were gathered, as well as included thecitation and reference for each procedure used in the methodology] Lines 95-104, 166-167, 119-123, table1.

Round 2
Reviewer 1 Report
Comments and Suggestions for Authors
Accept in present form
Author Response
Comments 1: [Accept in present form]
Response 1: [Thank you for pointing this out. Thank you for recognizing our revised manuscript]
